# Phenotypic Evaluation of Fire Blight Outbreak in the USDA *Malus* Collection

**Laura Dougherty** [1], **Anna Wallis** [2], **Kerik Cox** [2], **Gan-Yuan Zhong** [1] **and Benjamin Gutierrez** [1,*]

[1] Plant Genetic Resources Unit, United States Department of Agriculture-Agricultural Research Service, Geneva, NY 14456, USA; laura.dougherty@usda.gov (L.D.); ganyuan.zhong@usda.gov (G.-Y.Z.)

[2] Plant Pathology and Plant-Microbe Biology Section, School of Integrated Plant Science, Cornell University, Ithaca, NY 14850, USA; aew232@cornell.edu (A.W.); kdc33@cornell.edu (K.C.)

\* Correspondence: ben.gutierrez@usda.gov; Tel.: +1-315-787-2439

**Abstract:** Fire blight, caused by pathogen *Erwinia amylovora*, is a major disease in *Malus*. Biological, chemical and cultural controls are efficient to manage fire blight, while rootstocks, and host resistance can limit damages. During the 2020 season a naturally occurring fire blight outbreak occurred in the United States Department of Agriculture (USDA) *Malus* collection, providing a unique opportunity to evaluate the diverse collection for fire blight susceptibility. The *E. amylovora* strain in the collection was identified as streptomycin resistant and characterized as CRISPR (clustered regularly interspaced short palindromic repeats) spacer array profile, 41:23:38. Fire blight severity was assessed using two approaches: (1) Average severity percentage, where the number of infected shoots was divided by the total number of shoots for the east and west facing sides of the tree; and (2) cut severity rating, where the trees were visually assessed after fire blight removal for amount of tree removed. Overall, 1142 trees of 41 *Malus* species were assessed for average severity and 2525 trees of 48 species were assessed for cut severity. A subset of 667 trees were for average severity in June and July to understand the disease progression. The species and trees presented here, can provide insight for future genetic fire blight resistance studies.

**Keywords:** apple; evaluation; fire blight; genetic resources; *Malus*





## 1. Introduction

Apples (*Malus domestica*) are an important fruit commodity worldwide. In 2018, the United States produced 9.8 billion pounds of apples valued at about 3 billion dollars [1]. Fire blight, caused by the bacterial pathogen *Erwinia amylovora*, is a devastating disease of *Malus* [2]. The first incidence of fire blight was recorded in the Hudson Valley of New York, U.S. in 1780 [2]. Since then, it has spread throughout the world. *E. amylovora* can infect blossoms, vegetative shoots, fruits, woody tissue, and rootstock crowns [3]. The disease can kill a tree or an entire orchard in a single season. It is estimated that fire blight causes over 100 million dollars of losses annually [4]. In 2000, a fire blight outbreak throughout Michigan in the U.S. resulted in approximately 400,000 tree deaths and 42 million dollars in losses [5]. Several top-ranking apples for production and consumer preference are highly susceptible to fire blight, including 'Gala,' 'Fuji,' and 'Jonagold' [3].

To manage fire blight, chemical, biological, and cultural controls are used [6,7]. Properly timed streptomycin treatments are one of the most effective means of managing fire blight [3,8]. Biological controls are able to manage fire blight to an extent but were not as effective as antibiotic treatments alone [8–10]. In addition to managing *E. amylovora* inoculum, other management approaches focus on host response. Prohexadione-calcium, trade name Apogee®, is an inhibitor of gibberellin biosynthesis, and effectively slows tree growth, while acibenzolar-S-methyl, trade name Actigard® stimulates natural tree defenses [3,4,11–13]. Streptomycin is the most effective management tool for control of

fire blight in North America east of the Mississippi River; however, streptomycin-resistant strains of fire blight have been recorded throughout the United States [14–17].

In addition to chemical controls, host resistance is highly desired to manage fire blight. Host resistance is a complex, quantitative trait, impacted by the underlying genetics of the tree, the strain of the *E. amylovora*, and environmental factors [18,19]. There have been numerous studies that have identified major and minor quantitative trait loci (QTLs) and genes response for host resistance in apple [20–24], including QTLs associated with specific *E. amylovora* strains [25]. Although resistance QTLs have been identified in *M. domestica*, such as on apple linkage group 7 in 'Fiesta' [26], they are predominantly identified in wild species. A major QTL on linkage group 3 was mapped in *Malus x robusta* 5 [27]; two QTLs on linkage group 12 were identified in 'Evereste' (crab apple) and *M. floribunda* 821 [20] capturing 40–70% of the variation; and major QTLs on linkage group 10 in *M. baccata* and linkage group 12 for *M. fusca* have also been reported explaining 50% and 85% of the phenotypic variation, respectively [28]. Additionally, a QTL on linkage group 12 of *M. arnoldiana* has been reported [29].

The United States Department of Agriculture (USDA) *Malus* collection in Geneva, NY is home to over 6000 apple accessions and 48 species and hybrids from around the world. During the 2020 growing season, a naturally occurring severe outbreak of *E. amylovora* swept through the USDA apple collection, offering a unique opportunity to evaluate the germplasm for fire blight susceptibility and resistance. Although observations of fire blight in the collection were documented from 1990–2000 incidences of naturally occurring shoot blight [30], the data were not systematically collected across a large number of accessions under comparable conditions. In this study, we identified a single *E. amylovora* strain, and recorded fire blight shoot severity for 1341 trees, totaling 48 species in the main collection and core collection. After pruning symptomatic trees, 2525 trees consisting of 48 species were rated for cut severity to infer fire blight susceptibility. Fire blight severity and cut severity were analyzed and significant differences were found in both between and within species. This data provides the first systematic evaluation and further insight into the diversity of fire blight tolerance in the USDA *Malus* collection, which will facilitate future exploration of the collection for breeding of new apple cultivars resistant to fire blight.

## 2. Materials and Methods

### 2.1. Plant Material

The USDA *Malus* collection is maintained in Geneva, NY, USA (42°89′50.19″ N, −77°00′68.31″ W). The permanent block (2830 accessions), referred to as the "main collection" in the present work, is grafted on EMLA7, fire blight resistant and semi-dwarfing (60–70% height) rootstock. A core genetic diversity collection of 236 accessions from the permanent block is grafted on B.9 (Budagousky 9) dwarfing rootstock. EMLA7-grafted trees are planted in duplicate 6 feet (1.8 m) apart, with the second tree removed after accession establishment. B.9-grafted trees are planted six feet apart and trellis supported. Rows are spaced 20 feet (6 m) apart. Accessions are maintained with conventional horticultural and pest management practices, including annual chemical applications and mechanical pruning [31]. Planting date varied from 1 to 20 years with an average of 14.6 years. *Malus* taxonomy is based on the accession information from the Germplasm Resource Information Network (GRIN-Global). These collections are contiguous and make up much of the *Malus* collection plantings.

### 2.2. E. amylovora Sampling, Streptomycin Sensitivity, and Strain Identification

Infected shoots from 47 randomly selected trees from the core and main collections representing the overall collection block were sampled on 17 June 2020, shortly after symptoms were detected. Samples were surface sterilized and cambium tissue was dissected and incubated on Crosse-Goodman medium at 28 °C for 48 h, as previously described [32,33]. Bacterial growths identified as *E. amylovora* based on their characteristic crater-like appearance were sub-cultured and DNA was extracted by suspending individual colonies in

sterile, deionized water and vortexing for 1 min. Presence of *E. amylovora* was confirmed using polymerase chain reaction (PCR) to amplify the plasmid pEA29 that is ubiquitous and unique to *E. amylovora* worldwide as previously described (Table S1) [33,34].

Sensitivity to streptomycin was determined by an initial phenotyping of growth of individual colonies on CG medium amended with 100 µg mL$^{-1}$ streptomycin as previously described [33,35]. Streptomycin resistance and mechanism of resistance were identified using PCR amplification of *strA* and *strB* genes as previously described by Tancos et al. (Table S1) [33]. This gene pair codes for an aminoglycoside transferase, one of the two known determinants of streptomycin resistance in *E. amylovora* [14].

Strain identity was determined by characterizing CRISPR (clustered regularly interspaced short palindromic repeats) spacer array profiles on a subsample of 11 isolates, haphazardly selected to represent the affected acreage, by PCR amplification and sequencing of CRISPR regions CR1, CR2, and CR3, as previously described [35,36] (Table S1). CRISPR spacer profiles of the 11 samples were determined by aligning them to reference sequences of known CRISPR array patterns using CLC Main Workbench v20 (Qiagen, Hilden, Germany) [35].

### 2.3. Fire Blight Severity Scoring

In late June of 2020, 1341 trees from the main collection were scored for fire blight severity (Figure 1). To assess the severity, the total number of first year shoots and number of first year shoots showing fire blight symptoms were counted for the east and west facing sides of a tree. The severity expressed as percentage was calculated for each side of the tree as follows where "FB" stands for fire blight and "1 yr" stands for first year shoots:

$$\text{FB severity } = \frac{\text{FB 1yr shoots}}{\text{Total 1yr shoots}} \times 100. \tag{1}$$

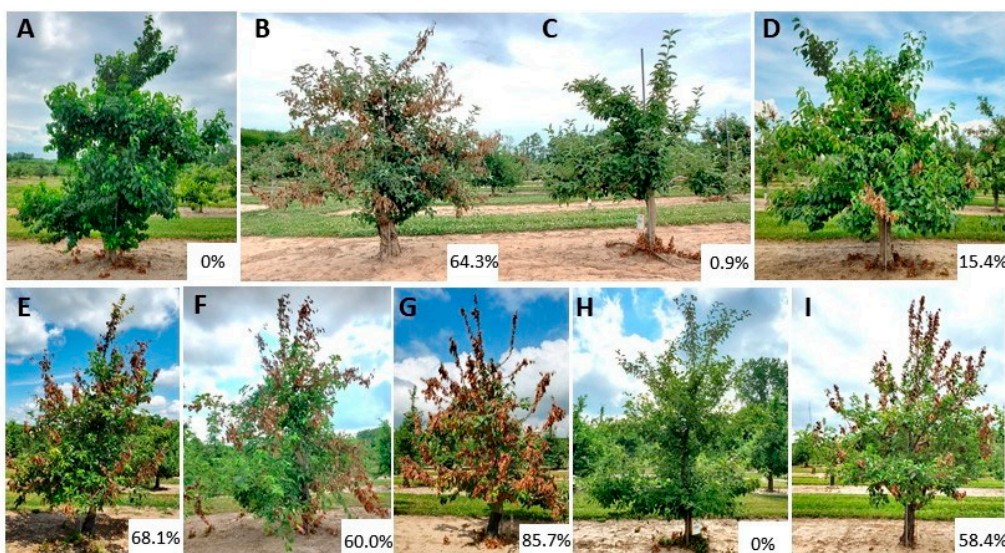

**Figure 1.** Examples of fire blight severity in the main collection. (**A**) Robusta 5 (*M. robusta*) (**B**) 'Ribston' (*M. domestica*) (**C**) 'Honora' (*M. domestica*) (**D**) 'Sugar crab' (*Malus* hybrid) (**E**) 'Royal Gala' (*M. domestica*) (**F**) KAZ 96 08-17 (*M. sieversii*) (**G**) 'Skryzhapel' (*M. domestica*) (**H**) GMAL 3173 (*M. baccata*) (**I**) 'John Standish' (*M. domestica*). The fire blight severity percentages are shown in bottom right of each panel. Pictures were taken during the initial fire blight scoring in late June 2020.

Severity assessments from the east and west were averaged to get the final tree severity. Approximately three weeks after the initial rating, 667 out of the 1341 trees from the main collection were assessed in mid-July for disease progression. The entire core collection was rated in early July, and a second time three weeks later in Late July to assess disease progression. To reduce bias for fire blight severity in small trees, we removed the samples

below the 15th percentile of total shoots. Trees containing fewer than 28 branches in the main collection and 22 branches in the core collection were excluded from analysis, leaving 1142/1341 trees from the main collection 199/236 trees in the core collection.

### 2.4. Fire Blight Cut Severity Ratings

As part of the routine management of the disease, fire blight infected shoots were systematically pruned, inhibiting an accurate evaluation of disease severity as described above. As such, a rating from "0" to "6" was used in the main collection to describe how much infected tissue was removed from the tree. Where "0" represents no visible cuts, "1" represents very light cuts, "2" represents light cuts, small branches removed, "3" light cuts, small and medium branches removed, "4" represents heavy cuts with large branches removed, "5" represents heavy cuts > 50% tree removed and "6" represents heavy cuts > 75% tree removed (Figure 2). Fire blight resistance was inferred based on cut severity, where "0" is likely resistant, "1", "2", and "3" are mildly resistant, "4" is moderately susceptible and "5" and "6" are highly susceptible. Small, immature trees were excluded from the analysis, including the samples below the 15th percentile from severity scoring. In total 2525 trees from the main collection were used for downstream analysis.

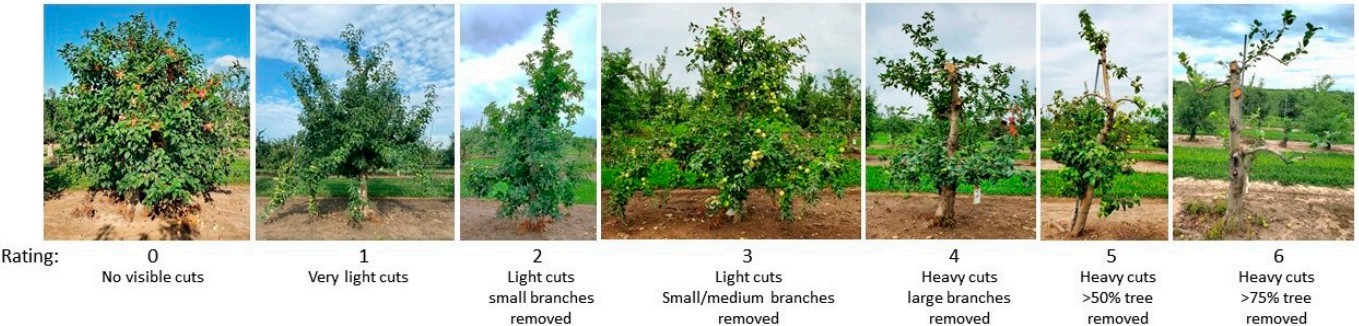

**Figure 2.** A visual representation of cut severity ratings with score and general description below each image.

### 2.5. Statistical Analysis

Statistical analysis was performed using ggstatsplot v0.6.5 [37] in R [38] and JMP® Pro, Version 15 (SAS Institute Inc., Cary, NC, USA). Statistical tests included Kruskal–Wallis, Dunn's multiple comparison test with Bonferroni correction and correlations.

## 3. Results

### 3.1. E. amylovora Strain Characterization

*E. amylovora* was successfully isolated from 40 of 47 samples; all had positive phenotypes for streptomycin resistance, indicated by growth on streptomycin-amended media at 100 µg mL$^{-1}$. For each isolate, the *strA* and *strB* gene pair was identified as the mechanism of resistance. The same CRISPR spacer array profile, 41:23:38, was identified for all 11 isolates collected over the affected acreage.

### 3.2. Fire Blight Severity

In the main collection 1142 trees consisting of 41 species were scored for fire blight severity in late June. The fire blight severity among all trees ranged from 0% to 85.7% (Figure S1). Only 11 of the 41 species had 10 or more trees. Of those 11 species, *M. domestica* had the greatest average severity of 27.3% (*n* = 534), followed by *M. toringo* (15.8%, *n* = 19) and *Malus* hybrids (14.9%, *n* = 136), while *M. ioensis* (5.4%, *n* = 19), *M. fusca* (4.0%, *n* = 20), and *M. baccata* (3.7%, *n* = 36) had the lowest severity. The majority of *M. sieversii* trees had average severity under 10%. Overall, *M. domestica, M. sieversii,* and *Malus* hybrids had the widest variations in severity with trees ranging from 0–85.7%, 0–79.3%, and 0–75.9% severity, respectively (Figure S2). A Kruksal–Wallis test was conducted for

species with 10 or more trees revealing significant severity differences between species ($p$-value = $3.65 \times 10^{-45}$). Dunn's multiple comparison test with Bonferroni adjustment resulted in eleven significant comparisons (Table S2).

In mid-July, 667 trees consisting of 34 species in the main collection were reevaluated for fire blight severity prior to pruning, approximately three weeks after the initial scoring of the 1142 trees (Figure 3). The overall severity of the subset ranged from 0% to 73.1% in June and 0% to 92.3% in July. When comparing the June and July severity, 19.3% of the trees had no change in severity; 36.4% had increased severity between 0.1 and 10%; and 3% had increased severity greater than 40% (Figure 4). On average, severity increased 12.7% in July. There were 113 trees in June with 0% severity. In July, 76 of those trees remained at 0% severity, while 34 increased to 0.4–10% severity and 3 trees increased to <10% severity but <20%. The largest increase in average severity (65.3%) was observed in a *M. domestica* tree that had 11.5% average severity in June and 76.8% average severity in July. Within the set of 667 trees, only 9 of the 34 species had more than 10 trees. For those, the highest severity in June was *M. domestica* (25.4%, $n$ = 354), *Malus* hybrids (12.4%, $n$ = 75), and *M. prunifolia* (8.0%, $n$ = 13) while the lowest severity was observed in *M. coronaria* (4.3%, $n$ = 12), *M. baccata* (1.08%, $n$ = 19), and *M. angustifolia* (0.17%, $n$ = 13). After the July assessment, *M. domestica* remained with the highest average severity (42.3%), followed by *Malus* hybrids (21.1%) and *M. sieversii* (13.6%), while *M. fusca* (6.0%), *M. baccata* (2.4%), and *M. angustifolia* (1.4%), had the lowest severity. In July, most of the *M. sieversii* trees had average severity of under 10% while *M. domestica* trees were more uniformly spread from 0–79.99% severity at 10% range intervals (Figure 5). The Kruksal–Wallis test was conducted on the June/July datasets with 10 or more trees per species revealing a significant severity difference between species in both June ($p$-value: $6.32 \times 10^{-42}$) and July ($p$-value: $2.92 \times 10^{-48}$). Dunn's multiple comparison test with Bonferroni adjustment was then completed, resulting in ten significant comparisons which were the same in June and July (Table 1).

In the core collection, 199 trees were assessed for fire blight severity twice, in early and late July (Figure S3). In the core collection there are 35 different species, however most species had one or two representative trees. Only 8 species had 5 or more scored trees, limiting species comparisons. Of those 8 species, *M. toringo* had the highest average severity (8.1%, $n$ = 6)) in early July followed by *M. domestica* (6.6%, $n$ = 55)) and *Malus* hybrids (4.4%, $n$ = 39). The lowest average severity was observed in *M. sylvestris* (0.25%, $n$ = 6)), *M. baccata* (0%, $n$ = 5)), and *M. orientalis* (0%, $n$ = 12)). In late July, *M. domestica* had the highest severity (9.3%), followed by *M. toringo* (8.72%) and *Malus* hybrids (6.48%) while *M. baccata* and *M. orientalis* remained the lowest with 0% severity.

Eighty-three trees were in the main and core collections on the EMLA7 and B.9 rootstocks, respectively. Typically, severity was higher on EMLA7 rootstock, with a mean difference in severity of 12.8% in June and 17.53% in July. Pearson correlations between accessions grafted on the two rootstocks were $r$ = 0.24 for June and $r$ = 0.29 for July datasets. There were 24 *M. domestica* trees with average severity of 10.9% in the core collection during the early July scoring and 32.0% in main collection for the late June assessment. The 19 *Malus* hybrids trees had average severity of 5.3% and 18.6% in the core and main collections while the 9 *M. sieversii* trees were 0.3% and 16.3%, respectively.

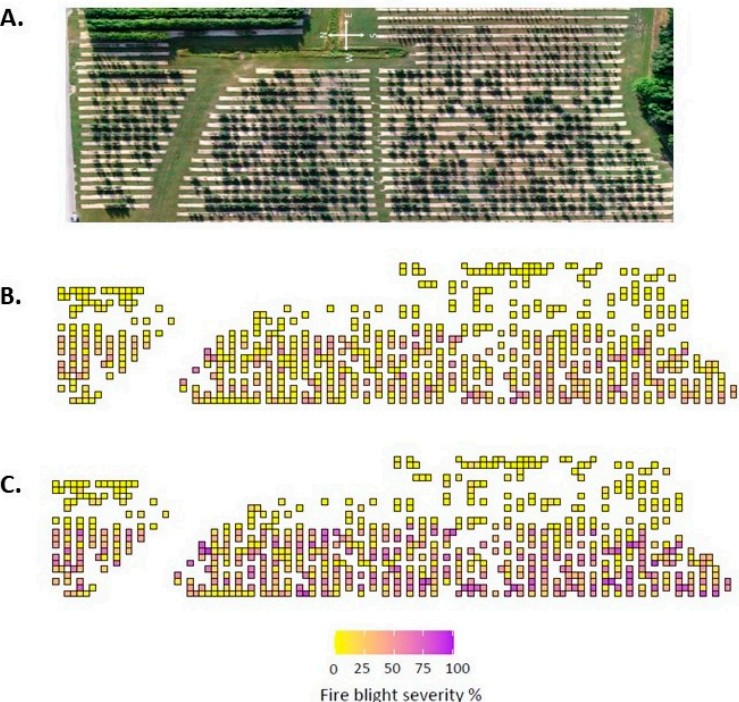

**Figure 3.** Heat map of fire blight severity for the 667-tree main collection subset. (**A**) Aerial image of the field. (**B**) Initial severity average in late June. (**C**) Severity average 3 weeks later, mid-July. Each square represents a tree. The compass in A. shows the cardinal directions of the field. Fire blight average severity calculated as (number of infected shoots/total number of shoots) × 100 for the west and east facing side of the tree, then averaged together.

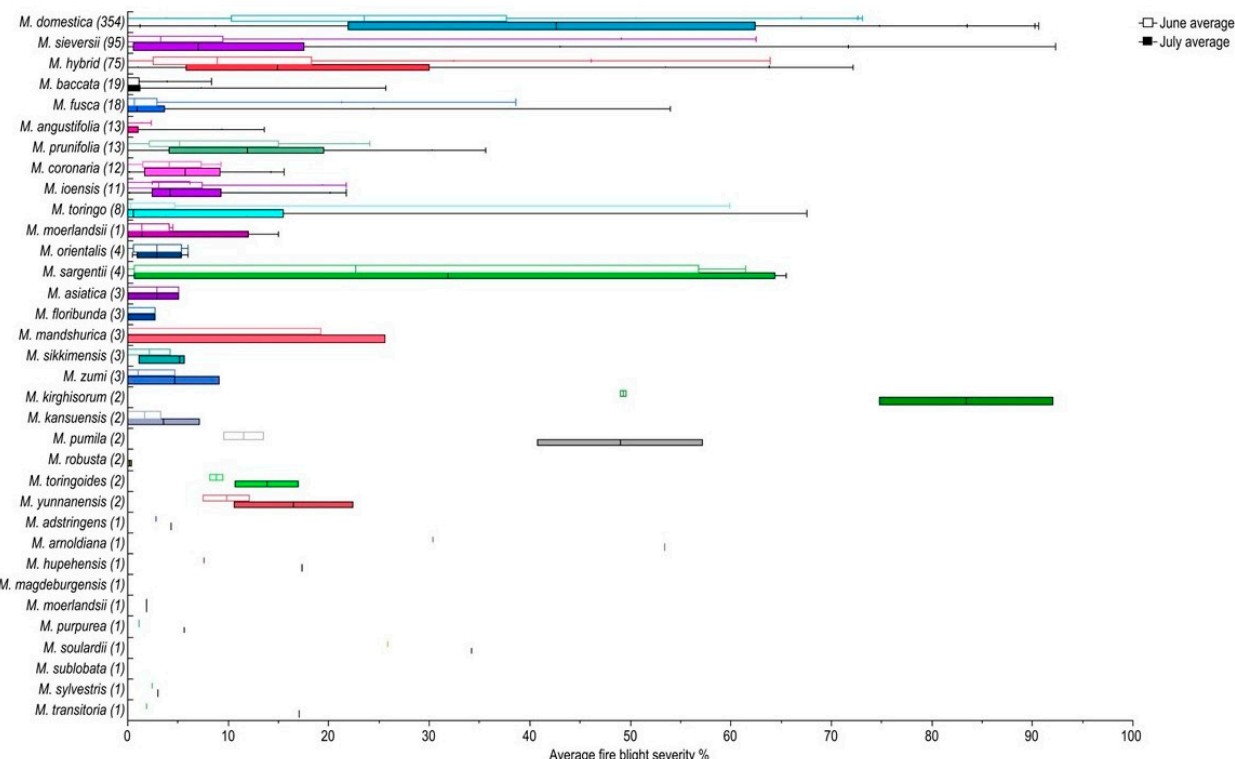

**Figure 4.** Quantile box plots of 667 main collection subset scored for fire blight severity in June and July. June is shown as open bars and July is shown as filled bars. The box indicates the 25th and 75th quantile and the median is shown as a colored line in the June box and a black line in the July box. The number of trees scored per species is shown in parenthesis. Species are ordered by descending tree count. Box plot colors delimitate species.

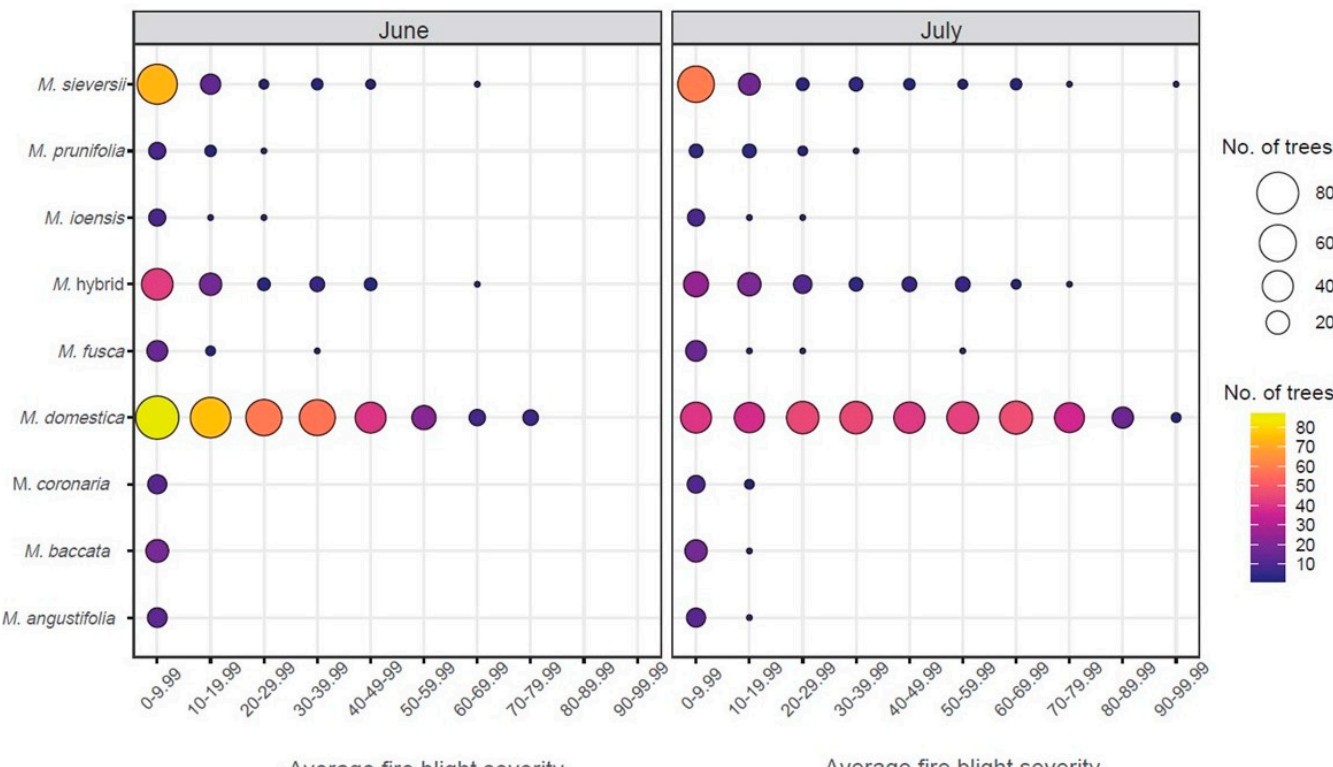

**Figure 5.** Main collection fire blight severity (%) in June (**left**) and July (**right**). The color and size of the circles indicates the number of trees. Only species with 10 or more trees are included in this figure.

**Table 1.** Significant pairwise comparisons of fire blight severity in the main collection 667 tree subset in June and July for species with 10 or more trees. Comparisons are listed with the highest severity first.

| Species Comparison | Average June Severity | June *p*-Value | Average July Severity | July *p*-Value |
|---|---|---|---|---|
| *M. domestica*/*M. angustifolia* | 25.42/0.18 | $6.08 \times 10^{-10}$ | 42.4/1.44 | $6.93 \times 10^{-10}$ |
| *M. domestica*/*M. baccata* | 25.42/1.08 | $1.26 \times 10^{-11}$ | 42.4/2.38 | $6.30 \times 10^{-13}$ |
| *M. domestica*/*M. coronaria* | 25.42/4.33 | 0.001 | 42.4/6.03 | $3.97 \times 10^{-5}$ |
| *M. domestica*/*M. fusca* | 25.42/4.48 | $1.02 \times 10^{-7}$ | 42.4/5.97 | $1.43 \times 10^{-9}$ |
| *M. domestica*/*Malus* hybrid | 25.42/12.43 | $7.78 \times 10^{-7}$ | 42.4/21.13 | $2.75 \times 10^{-8}$ |
| *M. domestica*/*M. ioensis* | 25.42/4.99 | 0.001 | 42.4/6.35 | $1.15 \times 10^{-4}$ |
| *M. domestica*/*M. prunifolia* | 25.42/7.97 | 0.019 | 42.4/12.51 | 0.002 |
| *M. domestica*/*M. sieversii* | 25.42/7.41 | $6.10 \times 10^{-22}$ | 42.4/13.63 | $2.09 \times 10^{-23}$ |
| *Malus* hybrid/*M. angustifolia* | 12.43/0.18 | 0.003 | 21.13/11.44 | 0.008 |
| *Malus* hybrid/*M. baccata* | 12.43/1.08 | 0.003 | 21.13/2.38 | 0.002 |

### 3.3. Cut Severity

Cut severity was scored for 2525 trees in the main collection. A heat map of cut severity ratings showed the heavy cut trees were randomly spread throughout the collection and not concentrated in clusters (Figure 6). To understand the distribution of cut ratings within species the frequency of the cut severity ratings was summarized (Figure 7). In total, 25% of the trees scored had cut severity ratings of four or higher indicating substantial cutting (Figure 2). There were 107 trees with rating of "6" and 75% of those were *M. domestica*. Additionally, 12.5% of the trees had a cut rating "0," of which only 15.5% were *M. domestica*. Twenty of the forty-eight species rated that had 10 or more trees and were used for analysis

(Figure 8). A Kruskal–Wallis test on cut severity data indicated that there were significant differences between species (*p*-value: $2.22 \times 10^{-58}$). Dunn's multiple comparison test with Bonferroni adjustment revealed fifteen significant comparisons (Table 2). Of those comparisons *M. domestica* (cut severity average 2.88) had a significantly higher cut severity rating than ten other species. *M. baccata* had significant differences compared to *M. coronaria*, *M.* hybrid, *M. ioensis,* and *M. toringo.* Lastly, *M. orientalis* and *M. toringo* were significantly different. Further observations highlighted species *M. floribunda, M. fusca, M. halliana, M. sikkimensis*, and *M. sylvestris* did not have any trees with cut severity ratings of "5" or "6" and much of the *M. orientalis* and *M. baccata* trees were rated between "0" and "2" (Figure 8A). Most of the *M. domestica* trees were rated between "1" and "4," while most of the *M. sieversii* were rated between "1" and "3" (Figure 8B).

The correlation coefficient between shoot severity and cut severity in the main collection subset was 0.66 in July. When comparing the 1142 June severity dataset, 667 June/July severity subset and 2525 cut severity datasets in the main collection, *M. domestica* was significantly more susceptible than *M. angustifolia, M. baccata, M. fusca, Malus* hybrids, and *M. sieversii* in all datasets. *M. coronaria*, *M. ioensis*, and *M. prunifolia* were significantly less susceptible than *M. domestica* in the large 1142 June severity dataset and 667 severity subset, but not in the cut severity data. *M. floribunda, M. halliana, M. orientalis, M. sargentii,* and *M. sikkimensis* were significantly less susceptible to *M. domestica* only in the cut severity rating data. Only *M. orientalis* met the 10-or-more tree requirement for the 1142 June dataset, but did not meet the requirement in the 667 subset, while the others did not meet the 10 or more trees in the 1142 or 667 subset. Additionally, *M. baccata* was significantly more tolerant than *Malus* hybrids in all three datasets.

There were 76 trees in the main collection 667 tree subset that had fire blight severity ratings of 0% in both June and July. Of those 76 trees, 55 were also rated "0" for cut severity, which included 17 species. The most represented species with 0% average severity were *M. sieversii* (*n* = 17), *M. baccata* (*n* = 9) and *M. angustifolia* (*n* = 5).

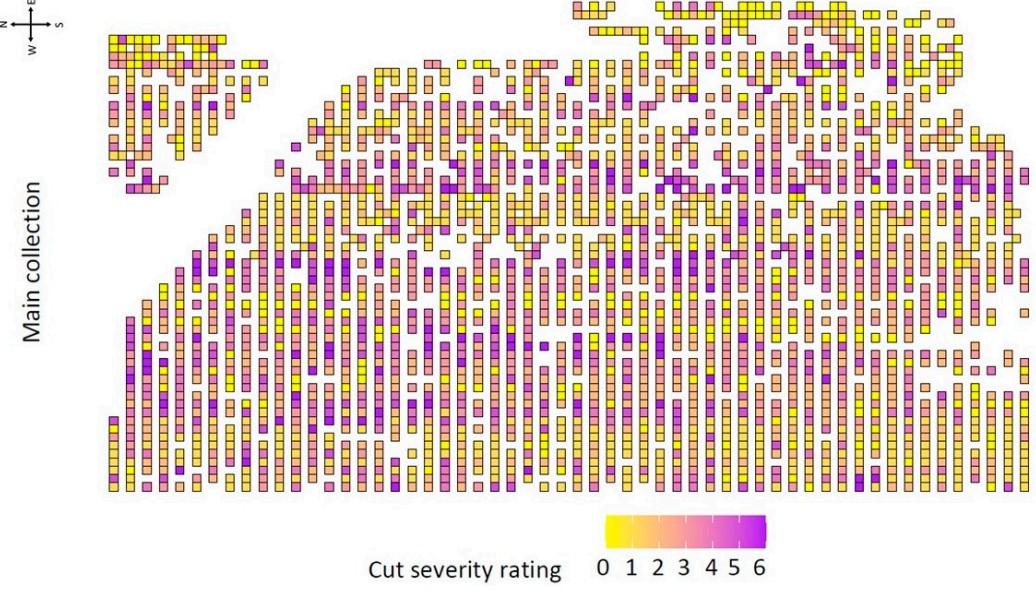

**Figure 6.** Heatmap of the main collection for cut severity ratings. Each tree is represented by a small box. Heavy cutting is observed throughout the orchard. White areas indicate no tree present. The cardinal directions of the field are shown in the top left corner.

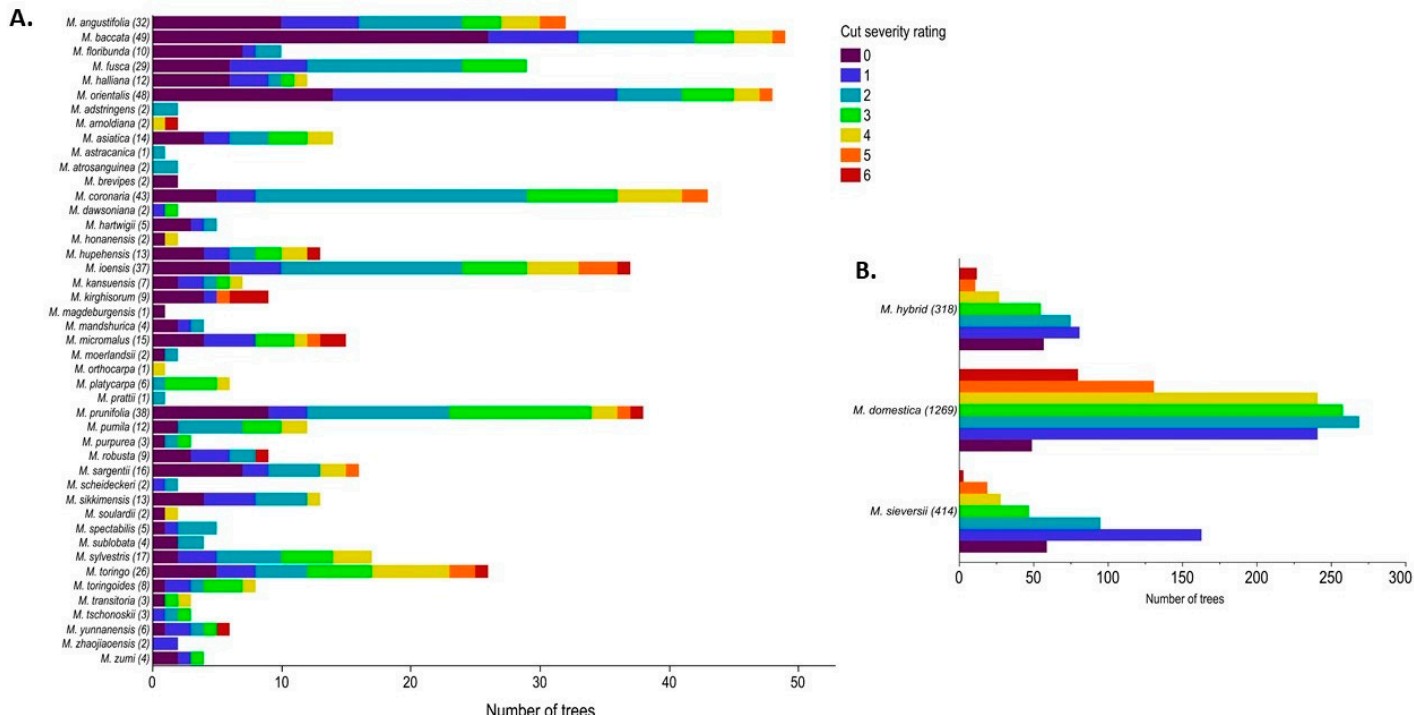

**Figure 7.** Cut severity ratings for the main collection. The number of trees per rating per species (**A**) Wild species (**B**) *Malus* hybrid, *M. domestica,* and *M. sieversii.* A rating of "0" represents no cuts. Ratings of "1", "2", and "3" represent light cuts while ratings of "4", "5", and "6" represent heavy cuts.

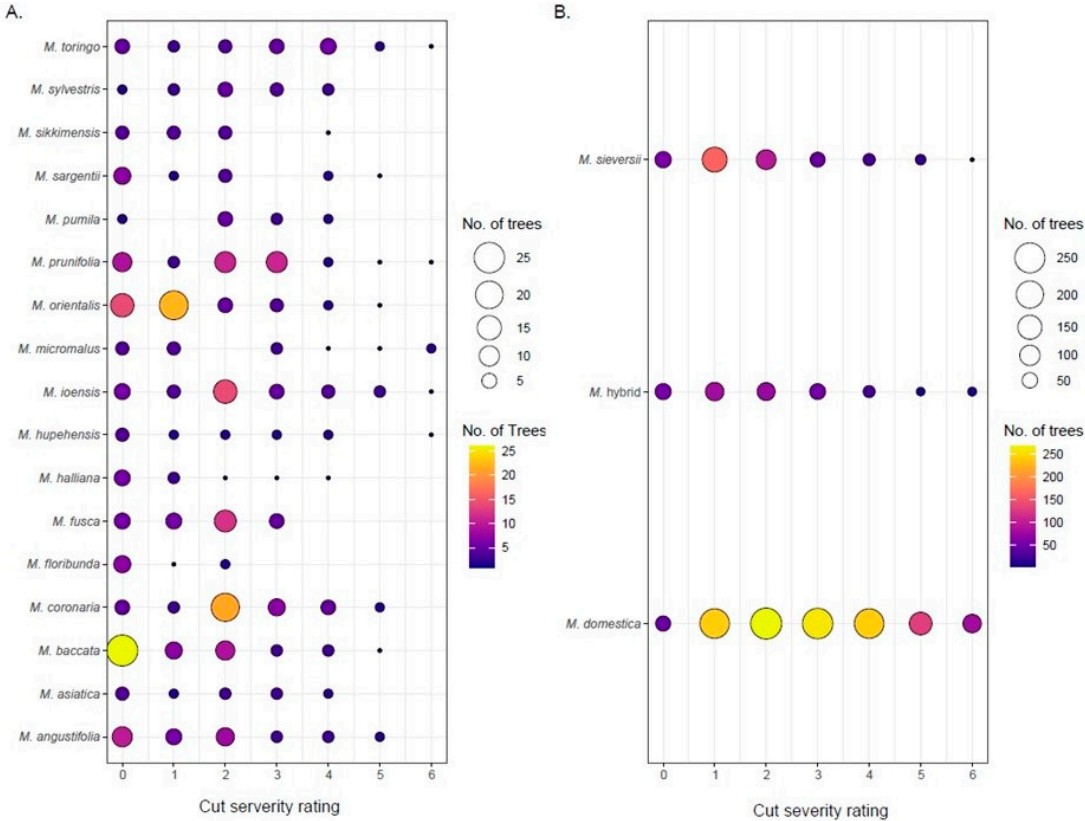

**Figure 8.** Main collection cut severity ratings with species containing >10 trees. (**A**) Wild species (**B**) *M. domestica*, *Malus* hybrid and *M. sieversii.* Circle sizes indicate the number of trees, as do the corresponding colors.

**Table 2.** Significant pairwise comparisons of cut severity ratings in the main collection between species with more than 10 trees. For each comparison, the species with the higher average cut severity rating is listed first.

| Species Comparison | Average Cut Severity Rating | *p*-Value |
|---|---|---|
| *M. coronaria/M. baccata* | 2.23/1.04 | 0.014 |
| *M. domestica/M. angustifolia* | 2.88/1.66 | 0.004 |
| *M. domestica/M. baccata* | 2.88/1.04 | $1.67 \times 10^{-13}$ |
| *M. domestica/M. floribunda* | 2.88/0.50 | $2.75 \times 10^{-4}$ |
| *M. domestica/M. fusca* | 2.88/1.55 | 0.004 |
| *M. domestica/M. halliana* | 2.88/1.00 | 0.006 |
| *M. domestica/Malus* hybrid | 2.88/1.98 | $6.01 \times 10^{-17}$ |
| *M. domestica/M. orientalis* | 2.88/1.19 | $1.92 \times 10^{-11}$ |
| *M. domestica/M. sargentii* | 2.88/1.44 | 0.049 |
| *M. domestica/M. sieversii* | 2.88/1.74 | $4.86 \times 10^{-35}$ |
| *M. domestica/M. sikkimensis* | 2.88/1.23 | 0.034 |
| *Malus* hybrid/*M. baccata* | 1.98/1.04 | 0.014 |
| *M. ioensis/M. baccata* | 2.27/1.04 | 0.038 |
| *M. toringo/M. baccata* | 2.54/1.04 | 0.012 |
| *M. toringo/M. orientalis* | 2.54/1.19 | 0.048 |

## 4. Discussion

The USDA *Malus* collection in Geneva, NY is one of the largest and most genetically diverse of its kind. The 2020 fire blight outbreak provided a unique opportunity to evaluate the genetic diversity of the collection to the disease. Random pathogen distribution is an assumption of natural fire blight observations in an orchard system. To a large extent, a random distribution pattern was observed in our study. Furthermore, we observed only one strain, CRISPR profile 41:23:28, responsible for the outbreak. This is the same profile as the first streptomycin-resistant isolates identified in NY State in 2002 [14]. The random distribution nature and single strain involved made our observations valuable for assessing the genetic diversity of fire blight tolerance across different species in the *Malus* collection. However, there are some inherent limitations in this study. Because of the tremendous diversity in our *Malus* collection, trees in the orchard can have varying blooming dates, which may compromise the comparisons of fire blight severity across different species. Moreover, the number of trees for some of the species are too few and we must exercise caution for making statistical inferences in these cases. Furthermore, we applied Apogee® for controlling the outbreak [39] during the season, which is intended to limit disease progression in the orchard and likely affected the severity evaluation; however, this treatment was applied uniformly to the block. Despite of these challenges, we believe that the data we collected served the purpose well in providing a baseline of fire blight susceptibility or resistance in the USDA *Malus* collection.

We used two complementary methods to evaluate the trees for fire blight damage; shoot severity and cut severity ratings. The correlation coefficient between fire blight shoot severity (%) and cut severity was 0.66 in July for the 667 main collection subset. Given the correlation strength, we can infer that cut severity ratings are representative of the fire blight infection. Although shoot severity is more informative, cut severity is more rapid and does not conflict with pruning to maintain tree health. Within the main collection there were numerous instances where severely infected trees were neighboring trees with no observable infection (Figure 3B,C). Those trees with no infection may have resistance to fire blight and should be further investigated. In a previous study evaluating susceptibility of apple cultivars to fire blight, trees were grafted on M.111 rootstock and inoculated with *E. amylovora* strain Ea153n in 2016 and 2017. Trees were classified as highly resistant, moderately resistant, moderately susceptible, and highly susceptible to fire blight based on the necrotic lesion divided by the total shoot length [40]. Despite differences in rootstock, fire blight strain, and method to calculate severity, we observed similar trends of cultivar responses to the outbreak. For example, in Kostick et al. [40],

'Arlet' was classified as moderately resistant; we observed severity of 5.9% and 7.8% in June and July respectively for this cultivar on EMLA7 rootstock. Similarly, 'Beacon' was classified as highly susceptible; we observed severity of 68% and 69% in June and July for this variety on EMLA7 rootstock [40].

As seen with severity, we observed heavily cut trees next to the neighboring trees with no cuts (Figure 6). Trees with ratings of "0," may be resistant to this strain of fire blight and those with cut ratings of "1" to "3" may be mildly resistant. In total, 12.5% of trees had no cuts and 62.7% of the trees had cut ratings of "1" to "3." Previous studies have determined that *M. domestica* accessions 'Liberty,' 'Empire,' 'Florina,' 'Novo Easygro,' and 'Cox's Orange Pippin' have fire blight resistance [40,41]. In our main collection, those accessions all had cut severity ratings of "2" or less indicating light or no infection. Additionally, in the core collection, 'Liberty' and 'Novo Easygro' had no fire blight incidence in either July assessment and the other listed accessions were under 2% shoot severity, except 'Cox's Orange Pippin' that had 5.95% severity in early July and 7.44% severity in late July.

Rootstock has a significant impact on fire blight resistance of the scion. The USDA *Malus* collection initially used M.9 rootstock but converted to EMLA7 following several fire blight outbreaks [30]. The core collection on B.9 dwarfing rootstock had far less fire blight incidence than the main collection, with an average severity of 4.6% and 6.2% for the June and July assessment, respectively. The dwarfing nature of the B.9 root stock and Apogee® treatments may have contributed to the reduced fire blight observations in the core collection. In previous studies B.9 rootstock has shown high survival rates when infected with fire blight [42,43]. We observed weak correlations between disease severity among the EMLA7 and B.9 rootstocks with higher fire blight severity on EMLA7 grafted trees. One notable exception was highlighted in *M. transitoria* PI 589384 with a severity of 83.8% and 17.1% on B.9 and EMLA7 rootstocks, respectively. Disease response × rootstock effects are likely confounded by random distribution of the pathogen and more controlled evaluations are needed.

Fire blight resistance QTLs have been identified in the wild species *M. arnoldiana*, *M. baccata*, *M. robusta*, *M. fusca*, and *M. floribunda* [20,28,29,44,45]. In this study we observed significant differences when comparing *M. domestica* to *M. baccata*, *M. fusca*, and *M. floribunda*. However, no significant differences between *M. domestica* and *M. robusta* or *M. arnoldiana* were observed and may have been due to low numbers of *M. robusta* and *M. arnoldiana* trees included in the analysis. The North American species *M. fusca*, *M. ioensis*, *M. coronaria*, and *M. angustifolia* were all significantly more tolerant to fire blight than *M. domestica* in at least two of the datasets, however *M. angustifolia* was the only one significantly more tolerant than *M. domestica* in all three datasets and could provide a new source of resistance deserving further evaluation. Previous studies have used limited numbers of *M. angustifolia* trees and to date no QTLs have been reported [46–48]. Additionally, *M. halliana*, *M. orientalis*, *M. sargentii*, and *M. sikkimensis* were all significantly more tolerant than *M. domestica*, based on cut severity ratings and may also be new sources for resistance although more severity data should be collected in the future. A report published in 1980 stated that five accessions of *M. halliana* were resistant to fire blight, but no additional information on how or why was provided [49]. The *Malus* collection contains four of the five accessions which had cut ratings of "0," "0," "1," and "2," although they were classified as *Malus* hybrids.

While most resistance has been discovered in wild species, the small fruit size and poor fruit quality present difficulties for gene introgression into commercial ready fruit [24]. In the collection we identified 49 *M. domestica* trees with no cuts and 59 *M. sieversii* trees with no cuts. These trees should be further investigated for potential sources of resistance. Previous studies have explored *M. sieversii* susceptibility to fire blight and found some accessions exhibit resistance [50,51]. *M. sieversii* trees have better fruit qualities than the other wild species and, given that they are a major progenitor species to the domesticated apple, they will be more suitable for breeding of commercial fruit [52,53].

This evaluation highlights the vulnerability of apple genetic resources to biotic and abiotic stress, while offering insight into fire blight resistance and susceptibility in the USDA *Malus* collection. Future preservation efforts for *Malus* should include advanced horticultural practices to mitigate disease pressure, but also leverage other approaches to safeguard genetic resources. During the 2020 fire blight outbreak, 326 accessions were severely diseased (cut severity "4" and above) and targeted for repropagation. Of those, 303 were also available as cryopreserved buds through the USDA-ARS National Laboratory for Genetic Resources Preservation in Fort Collins, Colorado, USA. Although field collections best facilitate access to woody plant germplasm, wild *Malus* seed banks and cryopreservation of scions safeguards allelic and clonal diversity against catastrophic loss. The streptomycin-resistant strain identified in the collection was never completely eradicated from orchards in Western New York. Prudent management and screening will be essential going forward to manage and prevent future outbreaks.

**Supplementary Materials:** The following are available online at https://www.mdpi.com/2073-4395/11/1/144/s1. Figure S1: Quantile box plot of fire blight severity for the main collection in late June 2020 by species. Figure S2: Main collection fire blight severity in late June in species with >10 trees. Table S1: Primers used for *E. amylovora* characterization. Table S2: Significant pairwise comparisons between species with 10 or more trees.

**Author Contributions:** Conceptualization, L.D., A.W., K.C., G.-Y.Z., and B.G.; data curation, L.D. and B.G.; formal analysis, L.D. and B.G.; investigation, L.D., A.W., and B.G.; methodology, L.D., A.W., and K.C.; visualization, L.D.; writing—original draft, L.D.; writing—review and editing, L.D., A.W., K.C., G.-Y.Z., and B.G. All authors have read and agreed to the published version of the manuscript.

**Funding:** This project was funded by the USDA-ARS Plant Genetic Resources Unit, Geneva, NY.

**Institutional Review Board Statement:** Not applicable.

**Informed Consent Statement:** Not applicable.

**Data Availability Statement:** Fire blight data will be available through the Genetic Resources Information Network (GRIN-Global) under the APPLE crop descriptors https://npgsweb.ars-grin.gov/gringlobal/descriptors.

**Acknowledgments:** We acknowledge USDA-ARS Northeast Area Summer Research Intern, Molly Dexter, for data collection. We are grateful to John Keeton, Bob Martens, Reece Perrin, Jackson Bartell, and David Osborne for field support.

**Conflicts of Interest:** Authors declare no conflict of interest.

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
