# Peer review of "Phenotypic Evaluation of Fire Blight Outbreak in the USDA Malus Collection"

_agronomy, doi:10.3390/agronomy11010144_

Round 1

Reviewer 1 Report

Comments to the authors

The focus of the manuscript 'Phenotypic evaluation of fire blight outbreak in the USDA Malus collection’ was the evaluation of a Malus collection after a naturally fire blight outbreak.

The manuscript is nicely written and presents valuable findings for further detailed evaluation of Malus genetic resources with a resistance to fire blight.

Please see the comments in the manuscript.

Author Response

Thank you for your time and comments. We have made the suggested changes in our word document. Please see the attached PDF for our response to each comment. 

Reviewer 2 Report

Generally should be shorter and clearer in the structure, sometimes also the language is not appropriate and the article should be read accurately by a native speaker!

2,47-48 meaning of the sentence is unclear

introduction can be shortened, e.g. 2,62-86, because there are many things coming later again; e.g. plant material of the collection, …

3,96 orchards ?

126 were scored

296 assume

Author Response

Thank you for your time and comments. We have made the suggested changes in our word document, particularly your comments on the clarity of our writing.  Please see the attached PDF for our response to each comment. 
